# Intra-Rater and Inter-Rater Reliability Analysis of Muscle-Tone Evaluation Using a Myotonometer for Children with Developmental Disabilities

**DOI:** 10.3390/healthcare11060782

**Published:** 2023-03-07

**Authors:** Heeae Seo, Jeongseon Kim, Changseon Yu, Hyoungwon Lim

**Affiliations:** 1Department of Physical Therapy, Graduate School, Dankook University, Cheonan 31116, Republic of Korea; 2Department of Physical Therapy, Gangdong University, Eumseong-gun 27600, Republic of Korea; 3Department of Rehabilitation Medicine, Dongguk University Ilsan Hospital, Goyang 10326, Republic of Korea; 4Dankook University Disabled Rehabilitation Research Institute, Department of Physical Therapy, Dankook University, Cheonan 31116, Republic of Korea

**Keywords:** myotonometer, reliability, developmental disabilities, muscle tone

## Abstract

Assessing muscle tone is an essential component of the diagnosis, prognosis, and treatment planning of developmental disabilities (DD) in children and is of great help in developing clinical diagnosis patterns. The purpose of this study was to investigate intra-rater and inter-rater reliability using the myotonometer, which is an assessment tool to measure muscle tone in children with DD. This study included 26 children diagnosed with DD. Two physical therapists measured the children’s muscle tone using a myotonometer. For all the muscles measured, reliability was determined using the intra-class correlation coefficient (ICC), the standard measurement error (SEM), and the minimal detectable change (MDC). The intra-rater reliability for all muscles was excellent (ICC = 0.75~0.78), except for the biceps brachii (ICC = 0.68). The inter-rater reliability was also excellent for all muscles (ICC = 0.75~0.95), and the SEM and MDC showed small measurement errors. Therefore, the intra-rater and inter-rater reliability of measurements by the myotonometer was found to be good or excellent. This suggests that the myotonometer is a tool that can objectively assess muscle tone, and it can be utilized in clinical practice to quickly and conveniently measure muscle tone in children with DD.

## 1. Introduction

Due to advances in the development of the field of medicine, an increase in the survival rate of children with a low birth weight, preterm birth, multiple births (twins and more), and perinatal infections has been noted. This has invariably increased the incidence of developmental disabilities in children [1]. The Developmental Disabilities Assistance and Bill of Rights Act of 2000 (Public Law No: 106-402) clearly identifies developmental disabilities in children as functional limitations in three or more of the main areas of life activity (self-care, receptive and expressive language, learning, mobility, self-direction, independent living capacity and economic self-sufficiency) that are evident before 9 years of age [2]. Developmental disabilities are also defined as serious chronic disorders that result from physical or mental impairments, or a combination of both. These include developmental delays as well as impairments in cognition, motor skills, vision, hearing, language, and behavior [3,4]. In particular, abnormal muscle tone is a common symptom in these children [5,6]. Schwarz et al. reported that hypotonia resulted in feeding problems due to difficulties in suckling following a decreased oral motor function and pharyngeal dyskinesia [7]. Goo et al. and Sharp et al. argued that abnormal muscle tone causes an unbalanced movement pattern and that it is difficult to limit muscle movements, leading to decreased muscle strength, abnormal alignment, and delayed acquisition of motor skills [5,8]. Therefore, an accurate evaluation of muscle tone is essential for establishing a diagnosis, prognosis, and treatment plan, and an objective evaluation is necessary [5,9,10].

An objective evaluation tool is one that collects information on the degree of functional performance and developmental stages of children [11]. Generally, the modified Tardieu scale (MTS) and modified Ashworth scale (MAS) are widely used as tools for assessing muscle tone in clinical practice [12,13,14]. Among these two evaluation tools, MAS is the most commonly used. It is commonly used to evaluate the effectiveness of rehabilitation interventions for the treatment of convulsions in patients with spinal cord injury (SCI) or neurological disorders [15,16]. However, the reliability of MAS has been questioned in several studies [12,16,17,18,19,20,21]. In a study that investigated the reliability of MAS, passive range of motion (PROM), and MTS in 16 children with cerebral palsy, Mutlu et al. reported a low reliability for MAS [17]. Yam. et al. also showed a low reliability in children with cerebral palsy [18]. MAS was criticized for its inability to distinguish between increased muscle tone and stiffness of soft tissues and for the absence of a correlation between functional changes before and after treatment [19,20,21].

Myotonometers are non-invasive portable devices that quickly and easily measure the mechanical properties of soft tissues such as muscles and tendons [22]. Previous studies have proven the reliability, validity, and precision of the myotonometer [20,21,23,24,25,26,27,28,29,30]. In sports injuries [23,24], pain [25,26], musculoskeletal-system [31] and nervous-system disease stroke [19,20,21], Parkinson’s disease [27,28], and healthy people [22,29], it was found to be highly reliable. Myotonometers are easy to carry and use. However, no studies have examined the intra-rater and inter-rater reliability of using a myotonometer in measuring muscle tone in children with developmental disabilities. The purpose of this study was to investigate the reliability of using a myotonometer within and between raters. Through this, we sought to know whether it is a suitable tool for quickly and easily measuring the muscle tone of children with developmental disabilities in clinical practice.

## 2. Materials and Methods

### 2.1. Participants

The subjects of this study were 31 children who were diagnosed with developmental disabilities and were receiving treatment at D hospital, located in Daejeon city, South Korea. Of these, 26 children whose parents provided consent and who participated in the second measurement were recruited (Figure 1).

The following were included in the study:(1)Children diagnosed with developmental disabilities (under 18 years of age) [30].(2)Children without excessive stiffness that interferes with functional movement. (MAS ≤ G1+) [32].(3)Children who can understand and follow the therapist’s instructions.(4)Children who can maintain a supine position for at least 10 min.

The exclusion criteria were:(1)Orthopedic surgery within 6 months.(2)Botulinum toxin injection within 6 months.

This study was conducted with informed consent from the subjects and their parents and was approved by the Institutional Review Board of Dankook University (Approval No. 2020-12-015-001).

### 2.2. Evaluation Tool

MyotonPRO (Myoton, Ltd., London, UK) is the strain (creep, C), viscoelasticity (decrement, D), muscle tone (frequency, F), circular recovery rate (relaxation, R), and hardness of the external force and muscle contraction. It is a very useful tool in the biomechanical analysis of muscles that control body movement using various parameters [33]. The device is applied under constant preload (0.18 N) to pre-compress subcutaneous tissues, and it exerts a brief (15 milliseconds) mechanical tap at a pre-determined force (0.4 Newtons), followed by quick release, causing damped oscillations that are recorded by the testing probe. This device is portable, inexpensive, easy to use, and convenient for re-cording the biomechanical and viscoelastic stiffness of myofascial tissues relatively quickly [12]. The tool was positioned perpendicular to the surface above the muscle to be measured, slightly touching the subcutaneous superficial tissue, and then a light and fast mechanical impact was applied. The reduced vibration of the muscle was then recorded using an accelerometer at the end of the muscle-tone test, and the numerical value of the muscle parameter representing muscle tone and biomechanical characteristics was calculated (http://www.myoton.com (accessed on 1 January 2023)).

### 2.3. Measurement Method

In this study, two physical therapists (A and B), with more than 3 years of clinical experience, received sufficient training for this tool and participated in the study after using the tool for at least 1 week. The inter-rater reliability is a measure of consistent results, even if the person measuring is different. Inter-rater reliability was assessed by using two raters (A and B) on the same child at the same time (afternoon). The parameters were measured three times per second. Intra-rater reliability was re-measured under the same conditions but after a week, in order for the inspectors (A and B) to exclude the learning effect. In this study, the tones of the following muscles of the right upper and lower extremities were measured: biceps brachii, brachioradialis, rectus femoris, and tibialis anterior. In addition, the frequency of oscillations was characterized by muscle tone and calculated as follows: oscillation frequency (Hz) = 1/T, where T is the oscillation period in seconds [20,21]. When measuring the tone of the biceps brachii, the participants were in supine positions, elbows flexed at 10°–15° and forearms supinated with towels placed under their wrists. A measuring tape was used to locate the test landmark, which was the midpoint between the anterior aspect of the lateral tip of the acromion and the medial border of the cubital fossa. When measuring the brachioradialis, the test landmark was at two thirds of the distance from the lateral supracondylar ridge to the styloid process (when the elbow was extended and forearm pronated). When measuring the rectus femoris, raters were in a supine position with their hips in a neutral position and their knees fully extended. The test landmark was at two thirds of the distance between the anterior superior iliac spine and the superior pole of the patella. Measurements of the tibialis anterior were taken at two thirds of the distance between the lateral condyle of the tibia and the medial cuneiform (Figure 2). The inspectors were prevented from seeing each other’s measured values, and the results were not known during the measurement process. To ensure consistency in the measurement points, thereby increasing the reliability of the measurement, landmarks were labeled with a marker that is harmless to the human body [19,20,21]. When the coefficient of variation (CV) exceeded 3%, the set was measured again [34].

### 2.4. Statistical Analysis

All data collected were analyzed using SPSS version 25.0. The characteristics of the sample population were assessed using descriptive statistics. Reliability measures were divided into relative and absolute reliabilities. The intra-class correlation coefficient (ICC) was used to determine the intra-rater and inter-rater reliability of the evaluation tool (excellent ICC > 0.75; good = 0.74–0.40; poor < 0.40); the intra-rater and inter-rater reliability was averaged through a paired *t*-test (*p* < 0.005). Two physical therapists obtained measurements for the same child and repeated the same measurements a week later. The standard error of measurement (SEM), which measures absolute reliability, calculates the total measurement error across repeated measurements resulting from performance differences in the child as well as assessor and instrument variability;

SEM = standard deviation (SD) times the square root of 1 minus the reliability estimate.

SEM = SD × √(1 − ICC), where SD is the pooled standard deviation of the first and second measurements.

SEM can be used to calculate the minimal detectable change (MDC) to estimate the minimum amount of change required to exceed the measurement error. The 95% confidence level of the MDC was calculated by multiplying the SEM by 1.96 and by multiplying the resulting value by the square root of 2 [34,35,36].
MDC = SEM × 1.96 × √2

## 3. Results

### 3.1. General Characteristics of the Participants

The average age of the study participants was 10 years. The general characteristics of the study subjects were age, sex, diagnosis, and MAS. These are shown in Table 1.

### 3.2. Intra-Rater Reliability

Intra-rater reliability was expressed as ICC (2,1), a direct reliability measure for the 26 subjects, one week after primary and secondary measurements with the myotonometer. Those of the biceps brachii and brachioradialis were 0.68 and 0.75, respectively, and those of the rectus femoris and tibialis anterior were 0.78, and 0.75, respectively. Measurements for all muscles but the biceps brachii showed excellent reliability. The measurement for biceps brachii showed a good reliability (Table 2).

### 3.3. SEM, MDC of Intra-Rater Reliability

SEM values were 1.80 and 1.75 for muscles in the upper extremities (biceps brachii and bra- chioradialis) and 1.34 and 2.19 for muscles in the lower extremities (rectus femoris and tibialis anterior), respectively. MDC values were 4.98 and 4.85 for the muscles in the upper extremities and 3.71 and 6.07 for the muscles in the lower extremities (Table 3).

### 3.4. Inter-Rater Reliability

As a result of inter-rater reliability, ICC (2, k), a direct reliability measure, showed excellent reliability in all four muscles, with 0.78 and 0.82 for the biceps brachii and brachioradialis and 0.95 and 0.93 for the rectus femoris and anterior tibia, respectively (Table 4).

### 3.5. SEM, MDC of Inter-Rater Reliability

SEM values were 1.11 and 1.13 for muscles in the upper extremities (biceps brachii and brachioradialis) and 0.27 and 0.59 for muscles in the lower extremities (rectus femoris and tibialis anterior), respectively. MDC values for the muscles of the upper extremities were 3.07 and 3.13, and for the muscles of the lower extremities they were 0.74 and 1.63, respectively (Table 5).

## 4. Discussion

The purpose of this study was to investigate the intra-rater and inter-rater reliability of using a myotonometer to assess muscle tone. In addition, we tried to determine whether it is suitable as a tool to quickly and conveniently measure the muscle tone of children with developmental disabilities in clinical practice. The results showed that the myotonometer not only showed a high reliability in children with developmental disabilities but also had relatively low SEM and MDC values.

In this study, the intra-rater reliability of muscle-tone measurements in the biceps brachii and brachioradialis was 0.68 and 0.75, respectively, and those in the rectus femoris and the tibialis anterior were 0.78 and 0.75, respectively. Therefore, the results of this study showed an excellent or good intra-rater reliability. The reliability of the measurements in this study was higher than that in elderly individuals [35] but lower than that in stroke patients or healthy elderly individuals [29,34,36]. Thus, it was assumed that the difference between the studies may be related to differences in pathology. Individuals with a decreased muscle mass have a relatively higher percentage of subcutaneous fat than normal individuals, making the accurate measurement of the muscle tone difficult, resulting in an increased measurement error [35]. Bandini, Linda et al. and Murphy, N. A. et al. reported that children with developmental disabilities had a higher obesity rate than other children, which may have affected the measurement of muscle tone [8,37]. Aarrestad et al. reported that in the relaxation and spontaneous contraction of muscles of children with cerebral palsy, the biceps brachii showed a high reliability (0.82 to 0.99), and the medial gastrocnemius showed a high reliability of 0.88 to 0.99 [38]. Therefore, the results of the intra-rater reliability of this study suggest that the myotonometer is reliable in measuring the muscle tone of children with developmental disabilities.

As in the previous study, as a result of the measuring tone, stiffness, and elasticity of healthy adults and the elderly, the biceps brachii was 0.92 to 0.95, rectus femoris was 0.92 to 0.95 and 0.86 to 0.94, respectively [39], and in stroke, this study showed that the same four muscles (biceps brachii, brachioradialis, rectus femoris, and tibialis anterior) showed a high reliability of at least 0.75 [20,21]. It also showed a high reliability of the muscle tone for biceps brachii (0.74 to 0.99), as well as for medial gastrocnemius (0.84 to 0.99) in relaxed and spontaneous contractions in children with cerebral palsy [38]. As such, the results of this study showed excellent reliability, suggesting that this tool can be used to measure muscle tone in children with developmental disabilities.

The reliability of intra-rater and inter-rater measurements in this study differed slightly from those reported in most previous studies. This is probably because, firstly, the experimental methods are different. In previous studies with a high confidence, the measurement period was set at short intervals of 1 min to 2 days [20,34,36,38,40]. However, in this study, the learning effect was excluded by setting the measurement interval to one week. Secondly, there were repeated measurements. In this study, the probe touched the muscle to be measured repeatedly, so as to obtain a CV of less than 3%. These repetitive measurements can affect the degree of muscle tone in children with motor impairments [35]. However, in some highly reliable studies, measurements were made without setting a reference point for CV [10,38,41]. In the scientific literature, myotonometric reproducibility between qualified physiotherapists and novice users has also been evaluated. Agyapong−Badu et al. [39] demonstrated a good and excellent inter-rater reliability in novice physical therapists for three muscle-tone parameters in young and old healthy men [39]. In infant-cerebral-palsy studies, novice users’ reliability was also studied with relaxed and contracted muscles, with moderate to high results with some exceptions [40]. These studies suggest that myotonometric measurements provide objective data that are not influenced by the rater’s experience.

In general, passive resistance torque during muscle stretching using isokinetic devices is considered the gold standard for measuring muscle stiffness and quantifying muscle-tone disorders [42,43]. On the other hand, myotonometry provides an assessment by quantifying tissue displacement with respect to perpendicular compression force. Muscle stiffness quantified by isokinetic dynamometry is obtained by a longitudinally applied force to all the muscle-tendon and articular structures [44]. Other studies have revealed the relationship between myotonometry and other muscle-stiffness assessment tools. Ditroilo et al. [45] studied the influence of the position of the knee joint on the contractile and mechanical properties of the biceps femoris muscle by myotonometry and tensiomyography, concluding that myotonometry presented a higher sensitivity to detecting changes in the three myotonometric parameters (frequency, decrement and stiffness). In another study where muscle stiffness in Parkinson’s disease patients was evaluated, myotonometric parameters were shown to correlate with electromyographic recordings in the biceps brachii muscle. Thus, it is proposed that myotonometry could be an alternative method to assess the passive muscle stiffness of skeletal muscles [28].

There are few studies on standard error of measurement (SEM) and minimal detectable change (MDC). The smaller the SEM and MDC values, the more reliable the device [19]. Therefore, the low SEM and MDC values shown in this study suggest a high confidence in the measurement results [19]. The results of SEM and MDC in the present study can be used as reference data for myotonometers. They can also help clinicians and researchers to identify minute changes in the muscle properties of the four muscles (biceps brachii, brachioradialis, rectus femoris, and tibialis anterior) between repeated measurements in children with developmental disabilities [34].

The limitations of this study are as follows: First, the sample size used to investigate reliability was too small. Second, important variables that could affect the muscle tone, stiffness, and resilience were not controlled by surrounding temperature and body temperature and by treatment at facilities other than the treatment room or home. Third, BMI or skin-fold thickness data had to be considered at baseline because obesity could interfere with the non-invasive evaluation of tone by this instrument.

Therefore, the limitations of this study should be taken into consideration in future studies by increasing the sample size and controlling important variables that can affect obesity, muscle tone, stiffness, and resilience.

## 5. Conclusions

This study conducted a reliability analysis of the measurement of muscle tone using a myotonometer for children with developmental disabilities. The intra- and inter-rater reliabilities were good or excellent, as measured using a myotonometer in four muscles (biceps brachii, brachioradialis, rectus femoris, and tibialis anterior) in children with developmental disabilities. The results also showed low values in SEM and MDC. Therefore, the myotonometer is a tool that can objectively evaluate muscle tone and can be used to quickly and conveniently measure the muscle tone of children with developmental disabilities in clinical practice.

## Figures and Tables

**Figure 1 healthcare-11-00782-f001:**
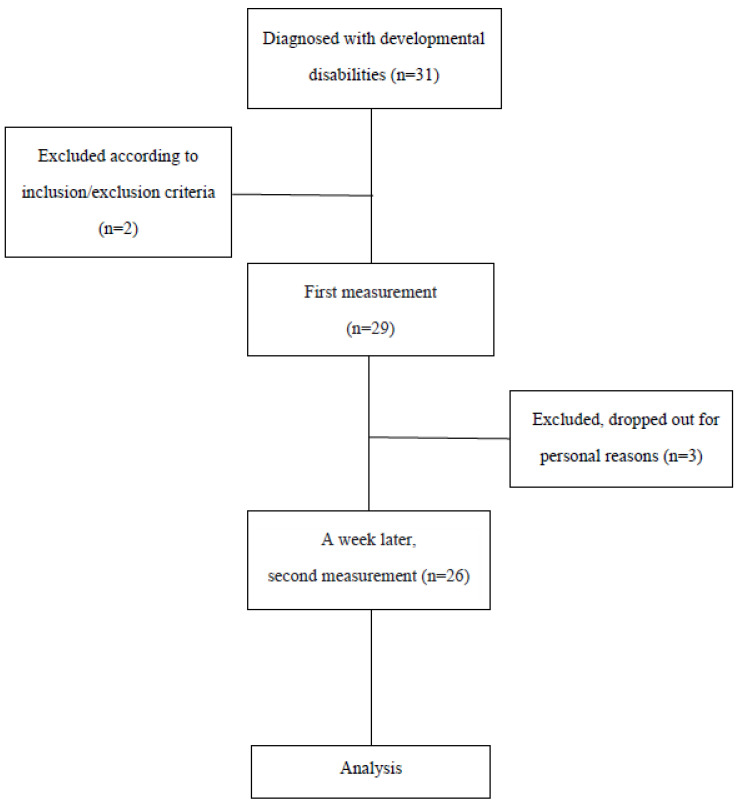
Flowchart of the study.

**Figure 2 healthcare-11-00782-f002:**
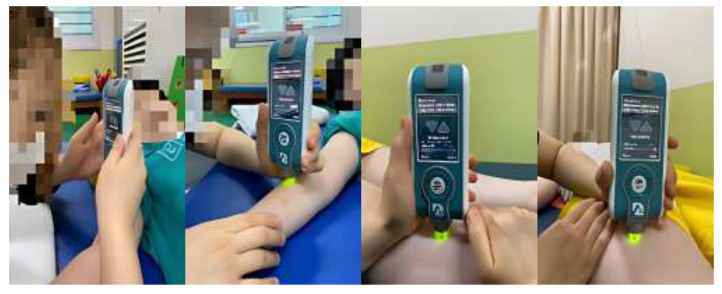
Muscle-tone measurements for each muscle were acquired using the MyotonPRO. Measurements of muscle tone.

**Table 1 healthcare-11-00782-t001:** General characteristics of participants (N = 26).

MAS	
G0(16) G1(6) G1^+^(4)	
Characteristic	
Age (Mean ± SD)	10.88 ± 3.2 (12.00)
Gender (male/female)	17/9
Diagnosis	N
Cerebral palsy	13
Unexplained developmental delay	6
Noonan syndrome	2
Lennox–Gastaut syndrome	1
Charcot–Marie–Tooth syndrome	1
Prader–Willi syndrome	1
William’s syndrome	1

MAS = modified Ashworth scale, ( ) = median (IQR) of age.

**Table 2 healthcare-11-00782-t002:** Intra-rater reliability of the myotonometer in the measurements of muscle tone.

		Intra-Rater Reliability			
Muscle (tone = Hz)	First Measurement	Second Measurement	ICC _(2,1)_	95%CI	*p*
Biceps brachii	30.15 ± 2.9	30.54 ± 3.5	0.68	0.29–0.85	0.003 *
Brachioradialis	34.47 ± 3.6	33.79 ± 3.3	0.75	0.46–0.89	0.000 *
Rectus femoris	29.53 ± 3.7	28.58 ± 3.1	0.78	0.51–0.90	0.000 *
Tibialis anterior	40.60 ± 4.4	41.52 ± 5.3	0.75	0.45–0.88	0.000 *

ICC: Intra-class coefficient, * *p* < 0.005.

**Table 3 healthcare-11-00782-t003:** Absolute Reliability Indices for the myotonometer’s muscle tone in intra-rater reliability.

	Intra-Rater Reliability	
Muscle	SEM	MDC
Biceps brachii	1.80	4.98
Brachioradialis	1.75	4.85
Rectus femoris	1.34	3.71
Tibialis anterior	2.19	6.07

SEM: Standard error measurement = SD × √(1 − ICC), MDC: Minimal detectable change = SEM × 1.96 × √2.

**Table 4 healthcare-11-00782-t004:** Inter-rater reliability of the myotonometer in measurements of muscle tone.

		Inter-Rater Reliability			
Muscle (tone = Hz)	First Measurement	Second Measurement	ICC_(2,k)_	(95%CI)	*p*
Biceps brachii	29.8 ± 2.6	30.7 ± 3.1	0.78	(0.51–0.90)	0.000 *
Brachioradialis	34.4 ± 3.3	34.2 ± 3.4	0.82	(0.59–0.91)	0.000 *
Rectus femoris	28.9 ± 3.0	29.5 ± 3.3	0.95	(0.88–0.98)	0.000 *
Tibialis anterior	41.3 ± 4.4	40.9 ± 4.3	0.93	(0.85–0.97)	0.000 *

ICC: Intra-class coefficient, * *p* < 0.005.

**Table 5 healthcare-11-00782-t005:** Results of the Absolute Reliability Index for myotonometer’s muscle tone in inter-rater reliability.

	Inter-Rater Reliability	
Muscle	SEM	MDC
Biceps brachii	1.11	3.07
Brachioradialis	1.13	3.13
Rectus femoris	0.27	0.74
Tibialis anterior	0.59	1.63

SEM: Standard error measurement = SD × √(1 − ICC), MDC: Minimal Detectable Change = SEM × 1.96 × √2.

## Data Availability

Data will be available from the corresponding author and will be released on reasonable request.

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
