# Peer review of "Intra-Rater and Inter-Rater Reliability Analysis of Muscle-Tone Evaluation Using a Myotonometer for Children with Developmental Disabilities"

_healthcare, 2023, doi:10.3390/healthcare11060782_

Round 1
Reviewer 1 Report
“Intra-rater and Inter-rater Reliability Analysis of Muscle Tone Evaluation Using a Myotonometer for Children with Developmental Disabilities”
Overall strengths of the article:
This manuscript explores intra-rater and inter-rater reliability using the myotonometer, which is an assessment tool to measure muscle tone in children with developmental disabilities. Abnormal muscle tone is the most common symptom in children with developmental disabilities. Therefore, an accurate evaluation of muscle tone is necessary for establishing a diagnosis, prognosis, and treatment plan. Myotonometers are non-invasive portable devices that quickly and easily measure the mechanical properties of soft tissues such as muscles and tendons. Several studies have proven the reliability, validity, and precision of the myotonometer. However, no studies have examined intra-rater and inter-rater reliability in measuring muscle tone in children with developmental disabilities using a myotonometer. This study aimed to investigate the reliability of using the myotonometer within and between raters. Through this, the authors sought to know whether the myotonometer is a suitable tool for quickly and easily measuring the muscle tone of children with developmental disabilities in clinical practice. The authors conducted a reliability analysis of the measurement of muscle tone using a myotonometer in 26 children with developmental disabilities. The intra- and inter-rater reliabilities were excellent, on the myotonometer in four muscles (biceps brachii, brachioradialis, rectus femoris, and tibialis anterior). Further, the low values in the standard measurement error, and the minimal detectable change, show that the myotonometer is a tool to objectively evaluate muscle tone. Therefore, the authors conclude that the myotonometer can quickly and conveniently measure the muscle tone of children with developmental disabilities in clinical practice.
Overall, this manuscript is very interesting and well-written, but I noticed some critical issues that are in the comments and need to be addressed.
Specific comments on weaknesses:
Major Critical Comments:
- The sample size is too small with no control and some conditions have only one patient.
- How much training is needed to use an myotonometer? Can the reliability be variable between a highly experienced clinician and a not so trained one?
Minor points:
1. Figure 2: Measurements of muscle tone. I could not get what the authors wanted to show in this picture. Some indications in the picture may help.
Author Response
- The sample size is too small with no control and some conditions have only one patient.
>>> We described the study's limitations as a problem with a small sample size. - How much training is needed to use an myotonometer? Can the reliability be variable between a highly experienced clinician and a not so trained one?
>>> We explained the points you raised based on previous research in the discussion section as follows.
Agyapong-Badu et [44] demonstrated good and excellent inter-rater reliability in novice physical therapists for three muscle tone parameters in young and old healthy men [44]. In infant cerebral palsy studies, novice us-ers' reliability was also studied with relaxed and contracted muscles, with moder-ate to high results with some exceptions [45]. These studies suggest that myotonometric meas-urements provide objective data that is not in-fluenced by the rater's experience. - Figure 2: Measurements of muscle tone. I could not get what the authors wanted to show in this picture. Some indications in the picture may help.
>>> We present the muscle tone measurement site by referring to previous studies using MyotonPRO. We revised it as follows:
"Muscle tone measurements for each muscle were acquired using the MyotonPRO".
Reviewer 2 Report
1. Baseline BMI/skin fold thickness data could have been collected and provided, as obesity may hinder non-invasive evaluation of tone by this instrument.
2. Kindly also provide median (IQR) of age, as only 26 patients were enrolled.
3. What was the developmental diagnosis of the LGS patient? LGS is, by definition, not associated with tone abnormalities.
4. Discussion: “In addition, we tried to determine whether 197 it is suitable as a tool to quickly and conveniently measure the muscle tone of children 198 with developmental disabilities in clinical practice.” Measurement of “quickness” (like time taken for measurement) or for “convenience” (like user’s or patient’s comfort level) is neither mentioned in Methods, nor provided in Results. This may be removed from the introduction and discussion.
5. Repetition of results in the discussion may be avoided.
6. Methods: Some information is repeated twice, which may be edited.
Author Response
- Baseline BMI/skin fold thickness data could have been collected and provided, as obesity may hinder non-invasive evaluation of tone by this instrument.
>>> We realized the problems pointed out by the reviewers and included the following in the study's limitations.
Third, BMI or skin fold thickness data had to be considered at baseline because obesity may interfere with non-invasive evaluation of tone by this instrument.
- Kindly also provide median (IQR) of age, as only 26 patients were enrolled.
>>> corrected - What was the developmental diagnosis of the LGS patient? LGS is, by definition, not associated with tone abnormalities.
>>> In our study, children with tonic seizures among the LGS seizure types were enrolled.
This child has stiffness in the arms, legs or torso.
- Discussion: “In addition, we tried to determine whether 197 it is suitable as a tool to quickly and conveniently measure the muscle tone of children 198 with developmental disabilities in clinical practice.” Measurement of “quickness” (like time taken for measurement) or for “convenience” (like user’s or patient’s comfort level) is neither mentioned in Methods, nor provided in Results. This may be removed from the introduction and discussion.
>>> Added to method. - Repetition of results in the discussion may be avoided.
>>> Deleted. - Methods: Some information is repeated twice, which may be edited.
>>> Deleted.